On the intrinsic sterility of 3D printing

Neches Russell Y. ryneches@ucdavis.edu 1
Flynn Kaitlin J. 2
Zaman Luis 3 4
Tung Emily 5
Pudlo Nicholas 2
1 Genome Center, UC Davis , Davis , USA
2 Department of Microbiology and Immunology, University of Michigan Medical School , Ann Arbor , MI , USA
3 BEACON Center, Michigan State University , East Lansing , MI , USA
4 Department of Biology, University of Washington , Seattle , WA , USA
5 Pivot Bio , San Francisco , CA , USA
Ahluwalia Arti
Electronic publication date: 2016 Dec 1
Publication date: 2016
Volume: 4
Electronic Location ID: e2661
Received 2014 Oct 3; Accepted 2016 Oct 5
Copyright: ©2016 Neches et al.
Copyright year: 2016
Copyright holder: Neches et al.
License: This is an open access article distributed under the terms of the Creative Commons Attribution License, which permits unrestricted use, distribution, reproduction and adaptation in any medium and for any purpose provided that it is properly attributed. For attribution, the original author(s), title, publication source (PeerJ) and either DOI or URL of the article must be cited.
License URL: https://creativecommons.org/licenses/by/4.0/

Keywords: 3D printing, Cell culture, Microbiology, Sterile technique, Methods, PLA, Polylactic acid, Pasteurization

Funding: Alfred P. Sloan Foundation AT&T Research Labs Fellowship BEACON Center for the Study of Evolution in Action University of Michigan Rackham Merit Fellowship Molecular Mechanisms of Microbial Pathogenesis Training Program NIH T32 AI007528 Russell Neches was supported by an Alfred P. Sloan Foundation grant to Jonathan Eisen. Luis Zaman was funded by an AT&T Research Labs Fellowship and the BEACON Center for the Study of Evolution in Action. Kaitlin J. Flynn was supported by a University of Michigan Rackham Merit Fellowship and the Molecular Mechanisms of Microbial Pathogenesis training program (NIH T32 AI007528). The funders had no role in study design, data collection and analysis, decision to publish, or preparation of the manuscript.

==============================
3D printers that build objects using extruded thermoplastic are quickly becoming commonplace tools in laboratories. We demonstrate that with appropriate handling, these devices are capable of producing sterile components from a non-sterile feedstock of thermoplastic without any treatment after fabrication. The fabrication process itself results in sterilization of the material. The resulting 3D printed components are suitable for a wide variety of applications, including experiments with bacteria and cell culture.

Introduction

Mass-produced, disposable products are ubiquitous in research laboratories. Roughly three billion microcentrifuge tubes are manufactured each year (Hashemi, 2006). The ubiquity of these products has helped to standardize molecular methods by reducing variability from experiment to experiment and from laboratory to laboratory. However, the proliferation of these products has come at the cost of in-house expertise in fabrication. Without these skills, researchers are increasingly dependent on vendors to anticipate and provide for their needs. If an experiment calls for a component that is unusual or unique, researchers are forced to improvise or to redesign the experiment using more readily available components. These restrictions are not necessarily detrimental; standardized materials are crucial for reproducibility. Nevertheless, there are experiments in which the need for a custom component cannot be avoided.

Many researchers have turned to 3D printing, a process by which three- dimensional objects are built up additively, to fill these needs. In some respects, the technology is more limited than traditional fabrication techniques used for laboratory equipment, such as metalworking or glassblowing; it is mostly limited to materials that can be melted and extruded at relatively low temperatures (150 °C–300 °C), such as thermoplastics. At the time of this writing, there are few inexpensive machines capable of combining more than one material. In other respects, 3D printing is more powerful than traditional fabrication techniques; additive manufacturing permits the creation of geometries that are impossible by other means, such as captured free moving parts. However, the principal advantage of additive manufacturing is the ability to move directly from a digital design to a finished part. It is not necessary to have a wide variety of specialized shop tools or the personnel and skills needed to operate and maintain them.

One of the most important properties of basic labware in the biological sciences is sterility, and one of the most frequent questions laboratory biologists ask when they first learn of 3D printing is, “Can I autoclave these things?” Unfortunately, most thermoplastics that are widely used in biomedical applications, particularly polylactic acid (PLA) and polyglycolic acid (PGA), will not survive a standard autoclave cycle (Rozema et al., 1991). Sterilization with γ-radiation is effective, but causes drastic changes to the biochemical properties of the material (Gilding & Reed, 1979).1 Here we detail our work demonstrating that the 3D printing process itself appears to be sufficient for ensuring sterility.

We note that the fused deposition modeling (FDM) 3D printing process, in which a thermoplastic filament is heated to melting and forced through a narrow tube under high pressure, resembles a sort of extreme pasteurization. Figure 1 compares the FDM 3D printing to several sterilization processes (note that thermal contact time is in log scale). The 3D printing process holds the material at a higher temperature for longer duration than both Ultra-High Temperature (UTH) pasteurization, which is used to produce shelf-stable milk (138 °C for two seconds) and high-temperature, short-time (HTST) pasteurization used for dairy, juice and other beverages and liquid ingredients (71.5 °C–74 °C for 15–30 s). The only legal pasteurization method that exceeds the thermal contact time typical of FDM 3D printing is mentioned in Title 21, Sec. 1240.61 of the Code of Federal Regulations, which permits milk to be treated at 63 °C for 30 min. This is a convenient sanitation regime for milk in non-commercial settings (indicated in  Fig. 1 as “stovetop” pasteurization). 3D printing is also both hotter and longer duration than thermization, a process used to extend the shelf life of raw milk that cannot be immediately used, such as at cheese making facilities.

Figure 1 Temperature and duration of various sterilization processes.

Temperatures and durations for various methods of sterilization compared to fused deposition modeling (FDM) 3D printing. The extrusion process most closely resembles pasteurization, in which non-sterile liquid is forced through a narrow, heated tube. High-temperature, short-time (HTST) pasteurization is used for milk, fruit juices and other beverages and ingredients. Ultra-high temperature (UTH) processing is used to produce products such as shelf-stable milk that do not require refrigeration. Stove- top pasteurization (30 min at 63 °C) is indicated as “stovetop” pasteurization. Thermization, a process used to extend the shelf life of raw milk that cannot be immediately used, such as at cheese making facilities. Typical autoclave cycles using prevacuum, and gravity displacement are indicated as “prevacuum” and “gravity,” respectively. A typical “flash” sterilization cycle for a gravity displacement sterilizer is also indicated. Pasteurization processes are indicated in black, autoclave processes in red, and thermization in orange.

For most materials and toolpaths, FDM 3D printing is also hotter than typical autoclave cycles for both gravity displacement steam sterilization and prevacuum steam sterilization. “Flash” steam sterilization using a gravity displacement sterilizer must reach 132 °C for 3 min. The Centers for Disease Control guidelines for gravity displacement steam sterilization require that the cycle reaches 121 °C for 30 min, or 4 min at 132 °C using prevacuum steam sterilization. 3D printing thermoplastics using FDM typically requires temperatures between 190 °C and 240 °C, depending on the material and the print parameters. Because the fabrication process calls for different extrusion rates over the course of a print, the thermoplastic generally dwells in the melt region of the nozzle for between ten seconds and several minutes.

To calculate the thermal contact time for an FDM 3D printer, we use the formula (1) Tf=mπdf22dnhf

where f is the feed rate in millimeters per second, h is the layer height, df is the filament diameter, dn is the nozzle diameter, m is the length of the melt zone. Because the length of the melt zone can be difficult to measure directly, but it may be inferred by using the area within the nozzle that has to be cleaned of melted plastic after a jam. For a feed rate of 50 mm/s, the thermal contact time in our 3D printer is about 16 s at 220 °C, although the print plan for a given part usually involves non-printing travel commands and regions where printing is carried out at a slower feed rate, resulting in longer thermal contact times.

Besides contact time and temperature, many sterilization protocols stipulate that high pressure must also be achieved. Depending on the protocol, pressures may range from about 40 to 220 kPa (6–31 PSI). The pressure inside the melt zone of a 3D printer nozzle is more difficult to calculate, as it depends on the fluid dynamics within the nozzle. Many common thermoplastics, such as PLA, are non-Newtonian fluids when melted, which further complicates the question. With those caveats in mind, we offer some rough estimates of the pressure within the nozzle.

At one extreme, the maximum possible pressure would occur when the force from the viscous fluid exiting the nozzle equals the maximum holding force of the stepper motor driving the extruder. For our printer, this is about 50–60 Newtons distributed over the area of the nozzle, which has a diameter of 0.4 mm. In principle, this would translate to a pressure of about 400,000 kPa (57,000 PSI) at the aperture, about two thousand times the pressure of an autoclave cycle. In practice, the holding force of the motor is distributed over a larger area by the hydrodynamics of the melted plastic. If the force were distributed over the whole inner surface of the nozzle (about one square centimeter), that would result in a pressure of about 600 kPa (87 PSI), or about triple the highest autocalve pressure. Normally, printers operate at some fraction of maximum flow rate, and of course melting thermoplastic is not a simple fluid, and so the pressure is not distributed evenly. In our experiments, it is likely that the pressure was often below autoclave pressures.

Nevertheless, the glass transition for materials like PLA occurs very abruptly, with only a few degrees separating the solid and liquid phase. Lowering the print temperature by a small amount can lead massive increases in pressure. With some experimentation, it should be relatively easy to operate a 3D printer with nozzle pressures well in excess of autoclave pressures. For example, the control firmware could be modified to make small adjustments to the temperature to match the flow rate, or the user could specify the temperature and flow rates in the print planning software to maintain a minimum pressure in the nozzle.

Here we report our findings for a battery of culturing experiments conducted with 3D printed parts manufactured with consumer 3D printers. Several variations of sterile technique were tested; we printed parts onto surfaces treated with ethanol, onto flame-treated aluminum foil, and under UV light. Finally, we printed onto non-sterile carpenter’s tape, and then handled the parts with flamed forceps. To our surprise, all of these methods seem to be at least somewhat effective at producing sterile parts. We found that the resulting parts appear to be sterile under a wide variety of culture conditions known to enrich for a broad spectrum of microorganisms.

This work was carried out in three laboratories across the United States, with experiments coordinated and results shared openly using Twitter. Much of this correspondence is directly referenced by this manuscript so that readers may follow how the research actually unfolded. The two 3D printers used are installed at the UC Davis Genome Center and the BEACON Center at Michigan State University, and most of the culturing work was done at the University of Michigan Medical School. After the initial experiments at UC Davis (Fig. 2), researchers at Michigan State University independently developed variations on those techniques. When the initial results were reproduced, a battery of test parts were prepared using several variations on the technique and mailed to the University of Michigan for culturing. The work was conducted in this way in order to reduce the “in our hands” effect, so that we could be reasonably confident that others could successfully achieve the same results.

Figure 2 Preliminary experimental results.

Growth after 96 h at 37 °C in a shaking incubator. The beaker labeled (A) contains LB media inoculated with PLA plastic extruded from the printer nozzle at 220 °C. The beaker labeled (B) contains LB media inoculated with a segment of unextruded PLA plastic filament from the same spool. The beaker labeled (C) contains uninoculated LB.

Results

In all of experiments described, the material used for 3D printing was non-sterile polylactide, or poly-lactic-acid (PLA) filament sourced from suppliers that primarily serve the hobbyist market. This material was selected for a number of reasons. PLA is very easy to work with in 3D printing, with good layer-to-layer adhesion and very little shrinking or warping. It is also biodegradable, which is attractive for environmental reasons. PLA and related polymers are also known to be non-toxic, bio-compatible, and are widely used in medical applications, notably soluble medical sutures. When noted, UV treatment was carried out by placing the 3D printer inside a laminar flow hood equipped with a 15 watt germicidal florescent bulb (Philips, model G15T8). The bulb remained activated during the printing process, and completed prints were collected in a sterile dish exposed to the UV while other test parts were printed. As a result, the UV doses were variable but substantial.

Enrichment experiments

To assess the potential for contamination after printing, 10 mm diameter hollow cylinders were printed under a variety of conditions and incubated in several types of liquid media at different temperatures and under aerobic, microaerophilic and anaerobic conditions. Initially, cylinders from UC Davis and MSU were grown in lysogeny broth (LB) for 96 h. No growth was observed in the experimental tubes or in the negative control, but high turbidity was observed in the positive control. Throughout this study, positive controls were prepared by dropping cylinders onto the laboratory floor followed by retrieval using ungloved hands from underneath the refrigerator or a similarly inconvenient location.

After these initial experiments indicated no growth on the 3D printed parts, several more cylinders were printed. At UC Davis, test parts were printed onto flame-treated aluminum foil and transferred into conical tubes using flamed forceps. One group of cylinders was printed while the printer was situated on an open lab bench, a second group was printed in a laminar flow hood, and a third group was printed in a laminar flow hood under a UV lamp. In the process, an ample supply of positive controls were created inadvertently. At Michigan State, test parts were printed onto an ethanol-treated build platform.

Further growth assays were conducted with each group of cylinders in LB, nutrient-rich ACES-buffered yeast extract (AYE) (Feeley et al., 1979) and Terrific Broth in aerobic conditions at 37 °C and 30 °C, revealing no growth from UV treated parts up to seven days post-inoculation (Table 1). Growth was observed with one non-UV treated part at 96 h, which was determined to be contaminated with flora typical of human skin via selective plating and light microscopy. This was likely due to a handling mistake after printing. See ‘Identification of contaminating organisms’ for methods of identification.

Table 1 Summary of experiments conducted.

Experiment	Material	Part	Media	Δt	°C	Oxy.	Fab.	Cult.	Repl.	Result	
Preliminary (Neches, 2014a; Neches, 2014d; Neches, 2014e)	Orange PLA	blob	LB	96 h	37	+	UCD	UCD	1	–	
First trial (Neches, 2014b; Neches, 2014c)	Orange PLA	tube	LB	96 h	37	+	UCD	UCD	6	–	
Small vessel (Neches, 2014f)	Orange PLA	vessel	LB	96 h	37	+	UCD	UCD	1	–	
Terrific Broth (Flynn, 2014a; Flynn, 2014b)	Orange PLA	tube	TB	96 h	37	+	UCD	UM	2	–	
AYE Broth (Flynn, 2014b)	Blue PLA	tube	AYE	96 h	37	+	MSU	UM	1	+ (error?)	
First MSU trial (Zaman, 2014c; Zaman, 2014b)	Blue PLA	tube	LB	96 h	30	+	MSU	MSU	3	–	
Filament trial (Zaman, 2014e)	Blue PLA	filament	LB	96 h	30	+	MSU	MSU	4	–	
AYE Broth 2 (Flynn, 2014c)	Orange PLA	tube	AYE	48 h	37	+	UCD	UM	2	–	
Cell culture	Orange PLA	tube	RPMI-1640	6 d	37	+	UCD	MSU	1	–	
Swimmer plate (Zaman, 2014a)	Blue PLA	track plate	Soft LB agar	n/a	37	+	MSU	MSU	1	n/a	
Meat Broth, Anaerobic (Flynn, 2014d)	Orange & Blue PLA	tube	Meat broth	2w	37	–	UCD & MSU	UM	2	+ growth in non-UV at 2w	
Swimmer plate, redesign (Zaman, 2014d)	Blue PLA	3-track round plate	soft LB agar	48 h	37	+	MSU	MSU	1	+ (handling error)	
Printed on blue tape cleaned with etoh (Lewis, 2014b)	Orange PLA	tube	RCM	7 d	37	–	UCD	UCD, Mills Lab	2	–	

To test for the presence of anaerobic organisms, parts printed with and without UV were incubated in anaerobic conditions at 37 °C using two growth media, AYE and a custom chopped meat broth (CM Broth) (Hehemann et al., 2012). After seven days, no growth was observed in any tube except the positive control. After 14 days, a tube containing a sample that had been printed without UV became turbid. The positive control and cells incubated from the non-UV treated part were analyzed via 16S rRNA sequencing and found to contain bacteria associated with human skin (Supplemental Information 1).

The germinants sodium taurocholate and glycine, known to germinate Clostridium difficile and some Bacillus spores, respectively, (Wilson, Kennedy & Fekety, 1982; Sorg & Sonenshein, 2008) were added to Brain Heart Infusion (BHI) medium and incubated anaerobically with 3D printed parts for 28 days at 37 °C. Microscopy and plating revealed no germination of these types of spores at weekly examinations.

Cell culture experiments

Sterile cell culture is a requirement for a variety of biological research applications. The biocompatibility of PLA and PLA-copolymers has been studied in vitro since at least 1975 (Schwope, Wise & Howes, 1975) and in vivo since at least 1966 (Kulkarni et al., 1966). These materials have been used for sutures and surgical implants in humans since at least 1974 (Horton et al., 1974). More recently, there has been a shift towards using 3D printed scaffolds in combination with cell culture for tissue engineering (Bose, Vahabzadeh & Bandyopadhyay, 2013). If the 3D printing process is sufficient to create sterile scaffolds, researchers could create useful scaffolds without damaging them with heat, steam, radiation or chemical sterilization programs.

We performed a simple assay to ascertain if 3D printed parts are sterile under cell culture conditions. 3D printed parts that had been printed either with or without UV treatment were cultured with bone marrow-derived mouse macrophages for six days. Contamination was assessed by plating on LB and charcoal yeast extract thymidine (CYET) agar plates and examination under light microscopy. No evidence of contamination was found either in cells alone (Fig. 3A) or cells cultured with parts either printed with (Fig. 3A) or without UV (Fig. 3A) as judged by growth on agar plates and microscopy. Cell morphology and growth rate appeared to resemble the control cells grown in the absence of a 3D printed part and no visible contaminants were observed. Additionally, cells grown in the presence of 3D printed parts were competent for infection by Legionella pneumophila (data not shown). Cells appeared to grow normally immediately adjacent to the part, though the opacity of the 3D printed part prevented inspection for growth directly on the printed surface. Thus, 3D printed parts do not appear to contaminate or affect the growth of bone marrow-derived macrophages under these conditions.

Figure 3 Mouse macrophage growth in the presence of 3D printed parts.

Macrophages derived from mouse bone-marrow after incubation with 3D printed parts that had been treated with UV (A), without UV treatment (B), and treated with UV after handling and before incubation (C) and a control set of cells grown without 3D parts (D). Photos representative of three replicates in two independent experiments. Cell size, morphology and confluency were determined to be consistent across all experimental groups.

Motility assay

To demonstrate the utility of directly 3D printing sterile labware, we designed a simple four-well plate that could be used to assay bacterial motility (Lewis, 2014a). Each well is 70 mm long and 10.3 mm wide, and holds approximately 2.5 mL of liquid media. The four-well plates were removed from the build platform by gloved hand, and were kept sterile in an empty 100 mm Petri dish. We filled each well with 2 mL of 0.2% w/v LB agar and allowed them to solidify for approximately 20 min. Then, we spotted 2 µl of a bacterial culture that had incubated for twelve hours into three lanes, and left the fourth as a control for contamination. In this experiment, we used three bacterial strains, JW1183, BW25113, and REL606. The first two are from the Keio Collection of single-gene knockouts, and REL606 is an E. coli B strain that was used to initiate the E. coli Long-term Experimental Evolution Project (Lenski et al., 1991); JW1183 is a ycgR deletion, and BW25113 is the ancestral strain of the Keio Collection (Baba et al., 2006). The choice of the ycgR knockout was suggested by Chris Watters as a potential bacterial “superswimmer” (Waters, 2014). Indeed, using this 3D printed plate, we were able to identify a strong swimming phenotype of the ycgR mutant (Fig. 4). Contamination was not observed in the control wells from several plates printed on painter’s tape, abraded foil, or abraded and flamed foil that was used to wrap the part after printing and stored overnight before use. These results demonstrate that direct 3D printing of sterile parts is a viable and useful approach for applications that may require a non-standard part.

Figure 4 A 3D printed motility assay device.

A custom device for a motility assay fabricated using 3D printing. The device was found to be sterile without autoclaveing if contamination during post-fabrication handling is avoided.

Materials and Methods

Preliminary experiment

A sterile glass beaker containing roughly 20mL of LB media was placed under the nozzle of a fused deposition modeling (FDM) 3D printer. The nozzle was heated to 220 °C, and the extruder drive motor was driven forward until about 20 mm of polylactide (PLA) filament had been melted and expelled through the nozzle and into the beaker. A tangle of molten and cooled PLA detached from the nozzle and fell into the beaker. The mouth of the beaker was then covered with sterile aluminum foil. An unopened sterile beaker of LB was prepared as the negative control. A positive control was prepared with a length of un-melted PLA filament from the spool. The three beakers were placed into a shaking incubator at 37 °C for 96 h. The experiment, the progress and the reuslt were announced in real-time on Twitter to generate feedback and suggestions from the community, which sparked collaboration described in this paper (Neches, 2014a; Neches, 2014d; Neches, 2014e; Neches, 2014b; Neches, 2014c). No growth was observed in the negative control or the beaker inoculated with extruded material, and robust growth was observed in the positive control. Experimental setup and results were posted on Twitter as they occurred.

3D printing

The preliminary experiment seemed to indicate a potentially useful killing effect from the nozzle’s heat and pressure, and so a slightly more realistic assay was conducted. A simple model was created using the OpenSCAD (Kintel & Wolf, 2000–2004) modeling language consisting of a cylinder of radius 4 mm and height 10 mm (Fig. 5).

Figure 5 Design and fabrication of test parts.

A very simple model of a cylinder was created in OpenSCAD and exported in STL format. (A) The G-code toolpath visualization of test part in Cura. The slicing engine was set to a 0.4 mm wall width (equal to the diameter of the nozzle), cooling fans inactive, no infill, a top and bottom layer height of zero, and a spiralized “Joris Mode” outer wall. (B) Test parts were then 3D printed on abraided and flamed aluminum foil at 220 °C with a feed rate of 50 mm/s.

cylinder( r=4, h=10 );

The model was exported in Standard Tessellation Language (STL) format (Burns, 1993). The manifold was then converted into G-code commands (Thomas, Frederick & Elena , 2000) using Cura (version 13.12-test on Linux), using a wall width of 0.4 mm (equal to the nozzle diameter), cooling fans inactive, no infill, a top and bottom layer height of zero, and a spiralized outer wall (“Joris mode,” after Joris van Tubergen) to produce a small, open tube. The G-code was stored on a SD card and printed on an Ultimaker kit-based FDM 3D printer (standard, current firmware builds distributed by Ultimaker were used). A small patch of aluminum foil was lightly abraded with fine-grit sandpaper to improve surface adhesion properties, and flamed over a Bunsen burner until signs of melting appeared. The foil patch was then affixed to the build platform, so that the build area indicated in the G-code toolpath would be entirely within the untouched center of the patch. The G-code toolpath was also examined to insure that the nozzle would contact no surface except the build area on the foil. Printing was then initiated with a feed rate of 50 mm/s at 220 °C. Once printing was complete, finished parts were immediately removed from the build area using flamed forceps and transferred to culture tubes or conical tubes for storage and  shipping.

Independent reproduction of growth experiment on printed component

The experiment described in ‘3D printing’ was replicated at Michigan State University on a kit-built Ultimaker 3D printer modified with an E3D all-metal hot-end with a 0.4 mm nozzle. A cylinder was designed using OpenSCAD with a radius of 4 mm and a height of 12 mm. The model was exported in STL format and sliced with Cura SteamEngine 13.12. The cylinder was printed with a wall thickness of 0.4 mm, a feed-rate of 10 mm/second (the effective speed with the minumum layer cooling time set to 5 s), and a nozzle temperature of 225 °C. The print surface was prepared with 3M Scotch Blue painters tape, and was lightly wiped with ethanol before printing began.

Two printed cylinders were transfered to sterile glass tubes filled with 4 mL of LB media with flamed tweezers. A fragment of unused filament was used as a positive control, and an uninoculated tube was used as a negative control. Tubes were transfered to a shaking incubator set at 30 °C. No growth was observed after 24 h in any of the tubes with printed parts, while the unused filament contaminated the media. After two days, another cylinder was printed and incubated in LB broth. Again, after 24 h no growth was observed. None of the tubes with printed parts showed signs of growth after 96 h (Fig. 6).

Figure 6 Independent replication of results.

After 48 h, only the positive control (A) was contaminated. Printed cylinders in LB did not appear to contaminate the media (C and D). Tube (B) contains uninoculated media.

Terrific broth experiments

Printed cylinders from UCD and MSU were dropped into glass culture tubes with 3 mL of AYE or TB broth in independent experiments and transferred to a roller in a 37 °C warm room. After 96 h, one of the “no UV” tubes in AYE broth became turbid with a mixed population of bacterial growth as examined by microscopy and plating on CYET agar (Fig. 7). Repeated experiments did not yield growth for these parts, and so the contaminaiton was likely due to a handling error. No growth was observed for any parts grown in TB (Fig. 8).

Figure 7 Test on solid media.

A total of 10 µl of each AYE tube (positive control, PLA plastic and negative control) was struck out on Charcoal Yeast Extract solid media and incubated at 37 °C to grow for 24 h. Growth revealed that the PLA test part (A) appeared to contain a different bacterial species than the positive control tube (B). Media from the negative control part was also plated (C). Using a light microscope, both bacterial growths appear coccoid, with the yellow colonies forming clumps more often. Experiment was repeated with parts from UC Davis and Michigan State, plus controls. Contamination was not observed.

Meat broth experiments

Test parts were incubated for two weeks under anaerobic conditions at 37 °C in chopped meat broth (CM Broth) (Hehemann et al., 2012). A non-UV treated test part fabricated at UC Davis exhibited evidence of growth (Fig. 9). The contaminated media was plated on BHI+blood media and allowed to grow overnight (see Fig. 7), and 16S rRNA sequencing was performed on resulting colonies (see ‘Identification of contaminating organisms’).

Figure 8 Test in Terrific Broth.

3D printed parts from UC Davis with and without UV treatment (B and C) were suspended in sterile Terrific Broth supplemented with potassium salts, along with a negative control (A) and a positive control (D). After 24 h at 37 °C, no growth was observed for parts treated with UV. Tube (F) contains a part from Michigan State after 48 h of incubation in Terrific Broth, along with negative and positive controls (E and G). No growth was observed 96 hours after inoculation.

Figure 9 Test in anerobic conditions.

After two weeks in anaerobic chamber at 37 °C in “meat broth,” a non-UV treated part from UC Davis exhibited evidence of growth. All other parts were limpid, aside from the positive control. Contaminated media was plated on BHI+blood agar overnight (see Fig. 7), and 16S rRNA sequencing was performed on resulting colonies.

Cell culture

Sterility of 3D printed parts was assessed by incubating each part with bone marrow-derived macrophages from femurs of C57BL/6 mice (Jackson Laboratories) cultured in RPMI-1640 containing 10% heat-inactivated fetal bovine serum (FBS) (Gibco) (Swanson & Isberg, 1996). Microscopy was performed by culturing macrophages in plastic dishes with 3D parts for 6 days after the initial isolation from bone marrow in L-cell conditioned media and examining under light microscope. The University Committee on Use and Care of Animals approved all experiments conducted in this study (principal investigator Michele Swanson; protocol reference number PRO00005100).

Identification of contaminating organisms

Contaminated media (see ‘Meat broth experiments’) was streaked onto BHI agar plates (Fig. 7) supplemented with 10% defibrinated horse blood (Quad Five, Catalog No. 210; Ryegate, MT, USA) for colony isolation. Bacterial colonies with unique morphologies were picked into chopped meat broth and genomic DNA was extracted from a 1 mL cell pellet using Phenol:Chloroform and ethanol precipitation after bead beating. Nearly full length 16S rDNA was amplified using primers 8F and 1492R (Eden, Schmidt & Blakemore, 1991) and run on a 1% agarose gel to confirm amplification and size. PCR products were purified using the Qiagen MinElute PCR purification kit (Catalog No. 28006), quantified and bidirectional sequenced at the University of Michigan DNA Sequencing Core. Sequencing reads were analyzed using the DNASTAR Lasergene software suite (DNASTAR, Inc., Madison, WI, USA). Results were used to search the nr database (Pruitt, Tatusova & Maglott, 2005) to using NCBI’s BLAST online search tool determine the closest relatives. A total of three unique bacterial colonies were analyzed; two from a positive control and one from a non-UV treated 3D printed part. All three were 99% similar to their closest database hit, and found to be common skin associated microflora. The positive control yielded sequences related to Staphylococcus epidermidis and Propionibacterium acnes. Similarly, the non-UV treated 3D printed part was also a Propionibacterium acnes indicating that the bacteria present were likely introduced to the 3D parts post printing.

Bacterial strains, culture conditions and reagents

For AYE growth experiments, 3D parts were cultured on a rolling spinner at 37 °C in N-(2-acetamido)-2- aminoethanesulfonic acid (ACES; Sigma)-buffered yeast extract (AYE) broth supplemented with 100 µg/mL thymidine (Sigma) (Feeley et al., 1979).

Terrific Broth (TB) experiments were conducted on a rolling spinner at 37 °C in media containing yeast extract, tryptone and glycerol supplemented with 0.17 M KH2PO4, 0.72 M K2HPO4.

Chopped Meat Broth and BHI-Blood agar experiments were performed in a Coy anaerobic chamber (Grass Lake, MI) at 37 °C (Hehemann et al., 2012).

Anaerobic experiments were performed in anaerobic chambers from Coy Laboratories (Grass Lake, MI) in Brain-Heart Infusion broth supplemented with yeast extract (5 g/L). 0.1% cysteine and 0.1% taurocholate were added as germinants.

3D printed parts from UC Davis

The following materials were prepared in Jonathan Eisen’s laboratory at the UC Davis Genome Center and shipped to Michele Swanson’s laboratory at the University of Michigan. All printed parts were printed using Printbl Orange 3 mm PLA filament at 220 °C with a feed rate of 50 mm/s, using the same G-code files described in ‘3D printing’, and placed into sixteen 50 mL conical tubes using flamed forceps. The contents of the conical tubes was as follows :

• Test objects, printed under biosafety hood (10×)

• Test objects, printed under biosafety hood with UV (10×)

• Test objects, printed under biosafety hood with UV, then dropped onto no-sterile surface during handling (2×)

• Test object, printed under biosafety hood with UV (1×)

• Test object, printed under biosafety hood without UV, dropped during handling (1×)

• Empty, unopened conical tube

• Test object, printed under biosafety hood without UV (1×)

• Test object, printed under biosafety hood without UV (1×)

• Test object, printed under biosafety hood with UV (1×)

• Test object, printed under biosafety hood with UV (1×)

• Test object, printed under biosafety hood with UV, handled with ungloved hands (1×)

• Test objects, printed on open bench and left on lab bench overnight (2×)

• Unused Printbl Orange 3 mm PLA filament (3×)

• Unused Laywoo-D3 cherrywood 3 mm printable wood filament (3×)

• Unused Protoparadigm White 3 mm PLA filament (3×)

• Unused Printbl Crystal Blue 3 mm PLA filament (3×).

3D printed parts from Michigan State University

Several printed parts were prepared at Michigan State University and sent to the Michele Swanson lab at the University of Michigan. Cylinders were printed using the same G-code and parameters described in ‘3D printing.’ All printed parts from Michigan State University were printed using Ultimaker translucent blue PLA. Each part was removed from the printbed using flamed forceps and transferred to a sterile 15 mL plastic tube. The contents of the tubes was as follows:

• Test objects, printed on blue painters tape wiped down with ethanol (3×)

• Test objects, printed on abraded foil wiped with ethanol and flamed (3×)

• Unused Ultimaker translucent blue PLA filament (3×).

3D printing systems and materials

The 3D printing systems and materials used in this study are relatively inexpensive and available to the public. While it is likely that nearly any 3D printer that uses thermoplastic extrusion will perform similarly for these purposes, the exact devices and materials used in this study are available from the following suppliers:

• Ultimaker Original with v3 hot-end (UC Davis). https://www.ultimaker.com/pages/our-printers/ultimaker-original

• Ultimaker Original modified with a E3D hot-end (Michigan State University). http://e3d-online.com/

• PLA (Poly-Lactic-Acid) filament, Blue-Translucent, 0.75 kg. 2.85 mm diameter. https://www.ultimaker.com/products/pla-blue-translucent

• PLA filament, Orange, 1.0 kg 3 mm diameter (2.85 actual). http://diamondage.co.nz/product/pla-standard-colours.

Discussion

This work was inspired by the observation that, while most 3D printed products cannot be autoclaved, the extrusion temperatures typically used in 3D printing are significantly higher than temperatures used in most autoclave cycles. This led us to wonder if 3D printing is an intrinsically sterile process.

Sterility is a difficult property to judge due to the impossibility of proving a negative. In the experiments we have presented here, we endeavored to create advantageous conditions for growth for a reasonably wide range of organisms, and particularly organisms likely to be problematic for experiments in clinical microbiology, cell culture and molecular biology. We used the “richest” rich media available to us, and attempted to induce germination of spores under aerobic and anaerobic conditions. Of course, this is not exhaustive, and the culturing conditions used would not detect the presence of (for example) Sulfolobus or Methanococcus maripaludis. We did not perform culture-independent sampling, which would be of obvious interest. However, as a practical matter, we find that the printing process does indeed produce functionally sterile parts which should be suitable a wide variety of experiments.

While 3D printing is likely not the ideal method for producing all labware under all circumstances, there are nevertheless a wide variety of applications and settings in which the ability to produce small batches of sterile parts would be extremely useful. The ability to manufacture sterile parts on premises during extended fieldwork in remote locations can reduce logistical risks. Schools can print materials for student laboratory projects. Researchers in developing countries can reduce their reliance on costly imported disposable labware. Otherwise well-equipped laboratories can more cheaply obtain fully custom sterile components.

Our experiments indicated that there are several reasonable approaches to sterile technique, though we did not attempt to establish which among them is optimal. We anticipated much higher rates of contamination than were actually observed. In more than twenty incubations, we found only two contaminated parts. Based on plating, light microscopy and 16S rRNA sequence obtained from the culture and on the fact that other parts prepared in the same way failed to produce growth, it is likely that the part was contaminated after printing. These experiments are not intended to establish a quantitative measure of the rate of contamination characteristic of the process, but rather to demonstrate that sterile parts can be produced by direct 3D printing of non-sterile thermoplastic feedstock.

Future work

While fused deposition modeling printers are by far the most common, widely available and inexpensive printers at present, there are several other 3D printing technologies. For example, there are a number of technologies based on materials that undergo photopolymerization. We happened to have two machines available to us that use photopolymerization, an Objet Eden 260, which uses an inkjet-like print head and a UV lamp, and a Formlabs Form 1, which uses stereolithography. We performed a variation of our preliminary experiment using cylinders printed using these machines, and found they were also able produce sterile parts (Fig. 10).

Figure 10 Test with alternative 3D printing technology (Objet Eden 260).

A group of cylinders were printed on an Objet Eden 260. 24 cylinders were transfered directly from the printing plate to culture tubes by scraping them from the build plate with the open tube. Two cylinders were removed with an ungloved hand to act as positive controls. Tubes were incubated for 96 h at 37 °C in LB media, revealing one contaminated tube.

The mechanism of sterilization in for these technologies is likely to be very different from FDM devices. It is possible that cells are destroyed by radiation; the Objet machine repeatedly exposes the build surface to intense UV radiation, and the Form 1 uses a 120 mW, 405 nm (violet) laser. However, the more likely killing mechanism is chemical, as the cross-linking chemistry of many photopolymerization systems is driven by high concentrations of free radicals. Unfortunately, the chemical composition of the input material and the precise nature of the reactions is proprietary. Formlabs was kind enough to point us to the catalog of their supplier of raw materials, but we were not able to deduce the chemistry of their system from this information alone. It is our hope that researchers more familiar with these polymer systems will take up this question, and perhaps design materials for these printers that can be certified for manufacturing sterile parts.

Supplemental Information

Supplemental Information 1 16S Sanger sequence data for contaminating organisms

Click here for additional data file.

Table S1 Temperature and duration of various sterilization processes

The table from Fig. 1 is provided here to make it easier to access the data within the table.

Click here for additional data file.

Additional Information and Declarations

Competing Interests

Author Contributions

Animal Ethics

DNA Deposition

Data Availability

1 For a detailed review of these studies, we recommend the review by Athanasiou & Niederauer (1996).

Author Emily Tung works for Pivot Bio, a startup company affiliated with the QB3 incubator, which is itself affiliated with UCSF.

Russell Y. Neches, Kaitlin J. Flynn and Luis Zaman conceived and designed the experiments, performed the experiments, analyzed the data, contributed reagents/materials/analysis tools, wrote the paper, prepared figures and/or tables, reviewed drafts of the paper.

Emily Tung conceived and designed the experiments, reviewed drafts of the paper, designed the initial experiment.

Nicholas Pudlo performed the experiments, performed DNA sequencing and identification.

The following information was supplied relating to ethical approvals (i.e., approving body and any reference numbers):

Six- to eight-week-old female A/J and C57BL/6 mice were purchased from Jackson Laboratories. Mice were housed in the University Laboratory Animal Medicine Facility at the University of Michigan Medical School under specific-pathogen-free conditions. The University Committee on Use and Care of Animals approved all experiments conducted in this study.

Principal Investigator: Michele Swanson

Protocol: PRO00005100

Protocol Title: Legionella Pneumophila, A Genetic Probe of Macrophage Function

Approval Period: 10/2/2013–10/2/2016.

The following information was supplied regarding the deposition of DNA sequences:

Sequencing was for identification only. Results were nearly 100% identical to Propionibacterium acnes and Staphylococcus epidermidis. There are many such sequences on GenBank already. For example, NR_074675.1 and and JF769744.1 matched our sequences to within the error rate of the sequencing technology. Due to this profound lack of novelty, we decided not submit our sequence data to any databases. However, we will provide it as a data supplement and it consists of three Sanger sequences.

The following information was supplied regarding data availability:

The raw data has been supplied as a Supplementary File.

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
