# Peer review of "On the intrinsic sterility of 3D printing"

_PeerJ, doi:10.7717/peerj.2661_

## Round 0.1 · original submission · Minor Revisions

All reviewer comments are pertinent. In particular could the authors address the generality of the approach to 3D printing (FDM or others?) and materials (PLA? others? ). Also can they comment on how melt temperatures likely influence the sterility?

Reviewer 1 ·

Basic reporting

In this manuscript, entitled "On the intrinsic sterility of 3D printing" by Neches et al., the authors investigate the ability of melt extrusion 3D printing to, by its high thermal window for extrusion of most thermoplastics, intrinsically sterilize extruded material. The authors do a good job describing the goals of the work and show support of their hypothesis that extruded PLA filament is more sterile than non-extruded filament. Sequencing of resulting colonies was a great idea.

Nevertheless, the work could be improved in several ways:

1) a simple diagram (perhaps as figure 1a) explaining melt extrusion would help to indicate where heating and sterilization as observed is happening. The authors further reference high pressures in the nozzle, but provide no quantitative measure or characterization.

2) the window of 10-120 s for FDM is quite large. The authors should indicate sterilization times as a function of melt extrusion rate, since faster rates of extrusion may not heat the plastic for sufficient time to lead to full sterility, according to their model of sterilization through extrusion.

3) the authors equate consumer-grade FDM PLA plastic with medical grade polylactic acid. However, the authors do not provide evidence of the ease with which their commercial PLA can be degraded, and PLA filament manufacturers do not indicate what their exact formulation (such as MW or other polymeric additives) or non-polymeric additives (including coloring) are added to the filaments. The authors should carefully distinguish FDM PLA filament from medical grade polylactic acid or provide sufficient characterization of said PLA filament to prove equivalence. In particular, the colorings chosen or degradation products may lead to cytotoxicity.

4) Figure 3 needs a scale bar. Further, the referenced data of "cell size, morphology, and confluency" did not appear to be presented anywhere. Further, the definition of "intermediate UV" and indeed even "UV treated" did not appear to be quantified. The authors need to provide the light wavelength used and an estimate or quantitation of the fluence (typically given as mW/cm2) and time scale utilized for these sterilization procedures to allow replication by others.

5) There are many kinds of "3D printing" as referenced by the overly broad title, but the authors only investigate FDM. Thus, the title should be narrowed to fit the manuscript, perhaps as: "On the intrinsic sterility of melt-extrusion 3D printing".

Overall this is an interesting and important work and would be suitable for publication if these concerns can be sufficiently addressed.

Experimental design

see above.

Validity of the findings

see above.

Additional comments

see above.

Reviewer 2 ·

Basic reporting

the paper is well written and well organized. the methods adopted to perform the experiments are clear and well described and the results are clearly discussed.

Experimental design

the experimental designed is well described, i would suggest to perform other experiments changing the topology of FDM realized scaffold, because the topology of cylinder scaffold used is simple, it must be realised scaffolds with more complex topology where the presence of interconnected struts can alter the sterility of structure.

Validity of the findings

the results are really interesting but i'm not sure if the scaffold topology is more complex these findings will be valide. More experiments increasing the 3D topology should be performed.

Additional comments

the paper is well written and well organized. the methods adopted to perform the experiments are clear and well described and the results are clearly discussed.the results are really interesting but i'm not sure if the scaffold topology is more complex these findings will be valide. More experiments increasing the 3D topology should be performed.

·

Basic reporting

- The submission follows PeerJ's policies
- The article is written in English, it is clear but it has a couple (about 5) misspelling errors that should be corrected
- The article provides sufficient background
- The structure is fine
- Figures are relevant to the contents
- The text is self-contained and results are VERY relevant to the hypothesis

Experimental design

- In my understanding the research here presented is original
- The research question is clear
- The investigation is the result of the collaboration of several laboratories
- Methods can be replicated by other investigators, though I am missing information about the firmware versions of the machines used in to print the different parts
- The ethical committee from one of the participating universities is mentioned regarding the use of animals in the tests

Validity of the findings

- The data includes a series of control experiments, seems to be consistent and replicable
- I am not familiar with the field the authors work with (sterile lab-ware) or the tests they have used to evaluate how sterile their self-made tools are. I cannot evaluate this aspect
- The conclusions are properly stated and linked to the research question. In my opinion, the authors are modest about the possibilities this experiments are offering

Additional comments

Please add information about:
- (minor changes) Firmware used in the machines where you printed the parts, specially if there were any alterations from the original ones
- (minor changes) External ways to assess the machines are working according to the information they show in their displays like using an IR temperature gun to confirm the temperature on the extruder, etc. It is important to know how far you went in making sure the machines were working as expected
- (optional) It would be great if you added some more information about your future plans in this field. I would love to see a large-size experiment where you asked people to create your normalized test (I consider the cylinder is a normalized test) and send it for testing alongside a description of the machine they are using

---

## Round 0.2 · accepted · Accept

I have reviewed your rebuttal and revised manuscript and although it has been a long time since the original submission I believe the topic is still of interest and original, and that you have addressed the prior concerns.

To provide more clarity, I suggest that you consider changing the title to "On the intrinsic sterility of fused deposition based 3D printing"